# Adiposity in Depression or Depression in Adiposity? The Role of Immune-Inflammatory-Microbial Overlap

**DOI:** 10.3390/life11020117

**Published:** 2021-02-04

**Authors:** Oliwia Gawlik-Kotelnicka, Dominik Strzelecki

**Affiliations:** Department of Affective and Psychotic Disorders, Medical University of Lodz, Czechoslowacka Street 8/10, 92-216 Lodz, Poland; dominik.strzelecki@umed.lodz.pl

**Keywords:** depression, metabolic syndrome, non-alcoholic fatty liver disease, hypothalamic-pituitary-adrenal axis, inflammation, oxidative stress, microbiota

## Abstract

Some of the most common and debilitating conditions are metabolic disorders (metabolic syndrome and non-alcoholic fatty liver disease) and depression. These conditions are also exacerbated by the fact that they often co-occur. Although the exact mechanisms underlying such relationships are poorly known, antipsychotic medication and antidepressant use, diet and physical activity, and lifestyle factors are believed to play a role; however, their high co-occurrence rate suggests a possible pathophysiological overlap. This paper reviews several possible bases for this overlap, including hypothalamic-pituitary-adrenal axis dysregulation, immune alterations with chronic inflammation, and oxidative stress. While it is entirely possible that changes in the microbiota may play a role in each of them, interventions based on the implementation of dietary and other lifestyle changes, supplementation with prebiotics or probiotics and faecal microbiota transplantation have failed to achieve conclusive results. A better characterization of the above associations may allow a more targeted approach to the treatment of both depressive and metabolic disorders. The paper also presents several practical applications for future studies.

## 1. Metabolic and Depressive Disorders

Some of the most common and debilitating disorders worldwide are metabolic disorders (MDs), e.g., metabolic syndrome (MetS) and non-alcoholic fatty liver disease (NAFLD), and depression [1]. The one-year prevalence of depression is approximately 6% worldwide while its lifetime risk (single or recurrent episode) is three times higher [2]. MetS is estimated to have a global prevalence of about one quarter of the population [3]; however, as the disorder encompasses a wide range of anthropometric, clinical and metabolic abnormalities, including obesity, insulin resistance (IR), dyslipidaemia, and hypertension, it is difficult to present a consistent definition for use in epidemiology studies. Finally, MDs often demonstrate higher mortality due to their co-occurrence with depressive disorders (DD) [4,5].

A recent meta-analysis [6] found that individuals with depression demonstrated a 1.5 times higher risk of having MetS than the non-depressed population, as well as a 30% prevalence of MetS in depressed subjects [6]. In addition, very recent data from the Netherlands Study of Depression and Anxiety (NESDA) sample [7] found that individuals with atypical depression tended to demonstrate significantly higher inflammatory marker levels, body mass index (BMI), waist circumference (WC), and triglycerides, or lower high-density lipoprotein cholesterol (HDL-C) than persons with melancholic depression. Additionally, both obesity and MetS were found to be independently associated with significant depressive symptoms in a large, nationally-representative sample, and that participants with both conditions had a higher rate of depression than other groups [8]. Furthermore, individuals experiencing a current episode of major depressive disorder (MDD) are significantly more likely to have IR than non-depressed individuals [9].

NAFLD is a multisystem disease that is widely regarded as the liver equivalent of MetS, and is characterized by excessive hepatic fat accumulation [10]. It is the most common cause of chronic liver disease in Western countries. Similarly to MetS, most deaths among IR and NAFLD patients are attributable to cardiovascular diseases (CVD) [11].

Although the exact mechanisms underlying the relationship between conditions such as MDs and depression are poorly known, several hypotheses have been proposed. Firstly, antipsychotic medication use has been associated with a significantly higher prevalence of MetS, and augmentation with newer antipsychotics in non-elderly patients with depression was associated with a greater mortality risk than augmentation with antidepressants [12]. The use of antidepressants in general is uncertain [13]. Diet and physical activity has been found to explain 23% of the association between depression and MetS [14].

The other factors coexisting in metabolic and depressive disorders remain poorly known. However, the high co-occurrence rate of DDs and MDs suggests a possible pathophysiological overlap. Although the precise mechanisms remain unknown, it has been proposed that hypothalamic-pituitary-adrenal (HPA) axis dysregulation, immune alterations, inflammation, oxidative stress (OxS), autonomic nervous system dysregulation, IR, brown (thermogenic) adipose tissue (BAT), or microbiome alterations may play a role [1,5,7,15,16,17,18,19,20]. Future studies aimed at creating new potential prophylactic or therapeutic methods should therefore attempt to determine etiopathological overlaps between the above syndromes, and possibly identify a subpopulation of patients sensitive to microbiota interventions.

## 2. Etiopathologic Features in Common

To begin with, both affective and metabolic diseases demonstrate dysregulation of the HPA axis [21,22]. In depression, this axis is typically upregulated, resulting in impaired negative feedback of cortisol to the hypothalamus, pituitary and immune system. The resulting excess release of cortisol into the circulatory system has a number of effects, including elevation of blood glucose, impaired neurotransmitter action, and desensitization of cortisol receptors, leading to increased activity of pro-inflammatory immune mediators [21]. However, it is worth noting that atypical depression is associated with hypofunction rather than hyperfunction of the HPA axis. In individuals with MDs, especially those with an abdominal obesity phenotype and IR, primary neuroendocrine dysregulation mechanisms are believed to play a key role. This could be due to genetic predisposition and altered coping with some environmental factors, such as stressors, lifestyle or nutrition [22]. Interestingly somatic diseases and DDs both appear to demonstrate hypothalamic inflammation [23]. However, the exact pathophysiology of the dysregulation of the HPA axis in depression and MetS has not yet been revealed.

Although inflammation is a highly beneficial process when occurring as a response to a threat such as injury or infection [24], prolonged low-grade systemic inflammation is not [25]. Such conditions, labelled chronic low-grade inflammation (CLGI) or metaflammation, are believed to originate, at least partially, from abdominal white adipose tissue (WAT). In this sense, CLGI can be regarded as a form of civilization disease. Most modern chronic conditions may be associated with such a chronic inflammatory state and Chronic Low-grade Inflammatory Phenotype (CLIP) has been proposed [26]. Indeed, recent studies have confirmed that depression, anxiety and MDs are associated with CLGI. The condition is characterized by increased circulating pro-inflammatory cytokines, such as tumour necrosis factor-alpha (TNF-α) and interleukin-6 (Il-6), altered leukocyte population frequencies in the blood, and the accumulation of activated immune cells in tissues, including the brain [16,27]. Although far from conclusive, emerging evidence suggests that chronic inflammation in the central and peripheral immune system may mediate a subset of DD [28] commonly concurrent with obesity and metabolic diseases. This association is especially seen in atypical depression, where subjects show increased appetite along with symptoms of depression, which then correlate with elevated serum (TNF-α) and C-reactive protein (CRP) levels [7]. Moreover, it was found that depressed patients with both obesity and MetS showed the highest levels of systemic inflammation measured by CRP [8].

Several pathological processes associated with obesity may inhibit proper differentiation of brown fat, among them genetics, inflammation and OxS [29]. Although BAT appearance and correct function have been proven to be connected with reduced risk of obesity, type 2 diabetes, or CVD for several years, and animal studies have shown a link between BAT and improved glucose and lipid balance [30,31], their connection with mood regulation seems a new research target [32]. It has been shown that appearance and higher activity of BAT protects against stress-induced obesity and diabetes in mice and lower pre-stress activation of brown adipocytes is connected rather with the vulnerability to this phenomenon [33]. On the contrary, BAT may be involved in higher inflammation state (mainly through Il-6) following psychological stress and thus brown fat may be perceived as a stress-responsive endocrine organ [34].

However, the roles of specific immune and endocrine cells and cytokines in the relationship between MetS and depression still remains inconclusive. Moreover, research evaluating effectiveness of interventions on metabolic and inflammatory pathways should include “immuno-metabolic” form of depression [15,18].

A number of studies have indicated that a combination of inflammation and OxS paves the way for the development of metabolic [35,36] and depressive disorders [37,38,39,40]. OxS, the state of imbalance between the oxidative and anti-oxidative systems of cells and tissues resulting in the damage of cellular macromolecules, is influenced by abnormal total antioxidant capacity (TAC), antioxidants, free radicals, oxidative damage, and autoimmune response products. The presence of higher levels of OxS has also been associated with poorer antidepressant treatment response; in addition, antidepressant use has been found to be associated with a decrease in oxidative and inflammatory markers [39,40]. Importantly, elevated OxS in MetS, known to promote vascular inflammation, has been shown to be a major risk factor for CVD [35].

A clearer understanding of the mechanisms linking MetS, depression, the HPA axis, inflammation and OxS could generate new potential therapeutic targets or patient-specific strategies aimed at combating both metabolic and depressive disorders.

## 3. Microbiota as a Connecting Factor

Recently, there has been much interest in the role of intestinal microbiota in the pathophysiology of civilization diseases. The human body is home to many microbiota, which are believed to play a crucial role in overall health, including immune status. However, while the densest microbiota are found in the gastrointestinal (GI) tract, their full diversity remains uncharacterized [41,42]. Interestingly, diseases characteristic of Western civilization are often connected with depletion of the microbial communities forming the microbiota [43].

Although the GI microbiota has been shown to be an essential part of the bidirectional and complex gut-brain axis (GBA) [44,45], the autonomic and the enteric nervous systems, and the neuroendocrine and the immune system also play important roles. The brain influences the motor, sensory and secretory modalities of the GI tract, and in return, the gut influences the function of the brain, especially those regions involved in stress regulation [46]. This connection between gut and brain is exemplified by the comorbidity between GI and neuropsychiatric diseases [47,48,49]. The diversity of the gut microbiota appears to play a significant role in the occurrence of mood and anxiety disorders [50,51,52,53]: for example, Clostridia were consistently found to be reduced in number, or absent, in those with depression, independent of the presence of anxiety, while the numbers of Bacteroides may be more associated with the presence of anxiety, irrespective of the presence of depression. Changes in the gut microbiota population have been associated with the severity of depressive symptoms in clinical population [54,55]: These differences suggest that the distribution of gut microbiota could help clarify the underlying pathology of comorbid clinical presentation [56]. Additionally, the degree of intestinal integrity and levels of inflammation markers have been found to be associated with the severity of depressive symptoms and response to treatment [54]. However, little is known on how microbiota health is associated with specific features of DDs.

It has been demonstrated that intestinal microbiota function influences the etiopathogenesis of MetS and that dysbiosis is associated with abdominal obesity [57,58]. The gut microbiota of obese individuals has been found to be rich in Firmicutes and Actinobacteria, and is able to derive more energy from food and facilitate fat storage [59]. In addition, bacteria-derived products can induce adipose tissue inflammation [60].

The pathogenesis of NAFLD is also exacerbated by gut dysbiosis due to its influence on the accumulation of fat in the liver; dysbiosis is believed to promote the intestinal absorption of monosaccharides and accelerate hepatic lipogenesis. Furthermore, the gut microbiota is believed to indirectly regulate the development of NAFLD by influencing the availability of microbial metabolites such as short-chain fatty acids (SCFA) [61]. However, the evidence so far is rather inconsistent and further clinical studies are needed to address the composition, function, and role of intestinal microbiota in the prevention and treatment of MDs.

Increasing evidence suggests that an aberrant gut microbiota may be bidirectionally connected with disturbances in the HPA axis [62], chronic inflammation [47,63], and OxS exacerbation in tissues [64], as well as dysfunction in BAT and reduction in the browning process of WAT [65,66]. Therefore, it may serve as a link between MetS, depression and dysbiosis. Indeed, the GI microbiota can activate or attenuate the HPA axis through antigens, cytokines, prostaglandins, or some of the metabolites that cross the blood-brain barrier (BBB) [62]. Dysbiosis alters the permeability of the gut barrier, and products of the microbiota can induce neuroinflammation by increasing inflammatory cytokine production and altering BBB permeability [63,67,68]. Dysbiosis may also promote OxS peripherally and in the brain by interfering with the local level of radical species and the antioxidant system [64]. The gut microbiota may as well influence BAT activity and browning of WAT at multiple levels [69]. Additionally, it was found that supplementation with probiotics (live microorganisms which, when consumed in adequate amounts, confer a health benefit on the host [70]) can decrease the levels of inflammatory markers in healthy and sick individuals [71,72] and restore, directly or indirectly, the oxidative balance [73,74]. A better characterization of the above associations represents a critical step at phenotyping and can serve as a preface to longitudinal clinical studies, and a more targeted approach to the treatment of DDs and MDs (Figure 1).

As different individuals can have taxonomically-varied but functionally-similar microbiota, it is important to perform functional co-assessments of microbiota health [75]. One set of potential microbial function biomarkers are SCFAs, which are the most representative metabolites following anaerobic fermentation of fiber. SCFAs are multi-target substances and can modify the activity of a variety of cells of the gastrointestinal, endocrine, metabolic, immune, and nervous systems [76]; they are believed to promote the function of the BBB and cerebral and increase the central expression of neurotrophic factors [77]. Interestingly, SCFA depletion has been reported in MDD patients [78], and their administration was shown to alleviate symptoms of depression in mice [79]. There is also some evidence that SCFAs can play an important role in regulating metabolic and cardiovascular health, by supporting reduced obesity, increasing the level of satiety factors and improving insulin sensitivity [61,80]. However, although SCFAs may well play an important role in metabolic-mood interaction, this possibility has not yet been systematically explored.

## 4. Microbiota Interventions in Depressive and Metabolic Disorders

A number of studies on dysfunctional microbiota have examined the value of interventions based on introducing dietary and lifestyle changes, as well as supplementation with prebiotics (non-digestible carbohydrates that promote the growth of beneficial bacteria) or probiotics and fecal microbiota transplantation (FMT) [81,82]. Dietary fiber or fish oil intake has demonstrated prophylactic and therapeutic efficacy in a range of diseases with CLGI, as well as increased abundance of SCFA-producing bacteria. Recently, a number of studies have examined the potential of dietary polyphenols, which act as prebiotics and may influence microbiota prevalence. More importantly, anti-inflammatory bioactive phenolic acids derived from dietary polyphenols have been shown to pass through the BBB and hence, directly influence the brain [83,84]. The introduction of a healthy diet, as an intervention targeting the microbiota, is an established way of alleviating symptoms of metabolic disease [85] and is perceived as an adjunctive method in the treatment of depression [86]. Similarly, exercise and a regular routine appear to be associated with a harmonious bacterial ecosystem [87,88].

Prebiotic application has been associated with decreased fasting glucose, improved insulin sensitivity, and lipid profile, reduced inflammation markers and modulation of neuroinflammation [89,90]. It has been proposed that probiotics may act via a number of routes, including modulation of the immune system and antimicrobial substances, increasing competition against pathogenic microorganisms, enhancing the intestinal barrier function, and increasing the production of anti-inflammatory molecules and antioxidants [91]. Their value based on depressive or anxiety outcome measures has been demonstrated in recent meta-analyses [55,92,93]. It has been suggested that such microorganisms can serve as a new group of drugs named psychobiotics [94]. However, there is too little consistency among study results, especially in terms of anxiety [95,96,97] and strain-dependence, and there is a need for studies based on larger groups of clinical subjects. It appears that probiotic intake may ameliorate some of the clinical components of MetS [98,99,100,101] together with some of its associated inflammatory biomarkers [102,103]. Similarly, supplementation was found to lead to an improvement in clinical features and OxS biomarkers in patients with type 2 diabetes mellitus (DM2) [73] or polycystic ovary syndrome (PCOS) [104]. However, discrepant data regarding the health benefits of probiotics on metabolic diseases have been reported which may partially result from the heterogeneity of therapies and protocols. Finally, it is possible that FMT may be of value in the treatment of psychiatric disorders and MetS, but the data is limited [105,106], and further research with larger sample sizes and stronger scientific design is necessary. In addition, a growing number of medications have been found to affect the gut microbiota [107]. However, the development of reliable interventions based on modulating the activity of microbiota requires further intervention trials based on standardized experimental designs and data analysis methods.

Currently, treatments for both depression and metabolic diseases remain suboptimal for many patients, and further intervention options are eagerly sought. Whilst microbiota interventions may bring benefits to some individuals who do not fully respond to antidepressant medications, the target clinical sample is unclear. It is therefore essential for psychiatrists to cooperate not only with psychologists but also with general practitioners, dietitians and other specialists. On research grounds, it would be beneficial to create a scientific team working on complex relationships between mental health, metabolic disorders and the microbiota-gut-brain axis. Such a team would include a psychiatrist with nutritional competence, a specialist of microbiology and molecular genetics, a biostatistician specializing in microbiota evaluation, as well as a consulting dietitian, an internist and a psychologist.

The use of natural and safe compounds, such as diet, exercise, pro- or prebiotics, may be an effective way to target the composition of GI microbiota and metabolic functions. Such an approach would undoubtedly be a useful tool for preventing and treating abdominal obesity and its related conditions in patients with depression, and improving the associated mood symptoms.

## 5. Practical Applications for Future Studies

Several definitions of depression are applied in clinical and research practice [108]. According to the upcoming International Classification of Diseases (ICD-11), depressive disorders include single episode or recurrent depressive disorder, and dysthymic and mixed depressive and anxiety disorder (MDAD) (Table 1). The new diagnosis, underlying its impact on everyday functioning and quality of life, is anxious depression [109]. Additionally, MDAD has been included in the ICD-11 because of its significance in primary care settings and anxiety apparent overlap with mood symptomatology [110]. Therefore, the entire category of DDs may be included in eligibility screening protocols when designing studies of new potential therapeutic methods for depressive patients.

It is now agreed that the most useful and widely accepted description of MetS, and one that predicts a high risk of developing diabetes, if not already present, is the criteria set by the International Diabetes Federation (IDF; Table 2). The IDF definition and criteria address both clinical and research needs, providing a tool suitable for worldwide use [111]. In addition, non-invasive markers of NAFLD, aspartate aminotransferase to platelet ratio index (APRI) and fibrosis-4 (FIB-4) scores, appear an effective criterion for stratifying patients and identifying biomarkers in subpopulations sensitive to specific treatment. Previous studies have found these markers to be effectively stratify patients for liver-related morbidity and mortality, with comparable performance to a liver biopsy [112].

It appears that no single marker exists to detect subtle alterations of the HPA axis, CLGI, OxS or intestinal dysbiosis in depressive or metabolic diseases. On the contrary, evidence indicates the need for multiple endocrine, inflammatory and microbiotic parameters to be included in such testing.

In conclusion, intestinal microbiota function and composition, HPA axis function indicators, CLGI markers, and OxS parameters may all serve as potential bioindicators for identifying depressive subpopulations sensitive to add-on treatment with microbiota interventions. Such a trial, if successful, could establish easy-to-use biomarkers for clinical practice. Given the personal and societal cost of treatment of civilization diseases, the research would contribute to an achievable advancement in health care for millions of people.

## Figures and Tables

**Figure 1 life-11-00117-f001:**
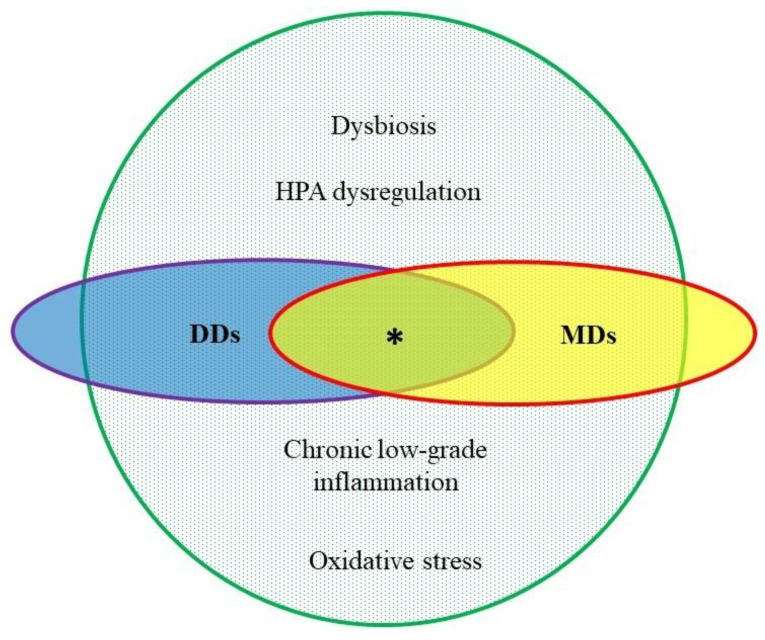
The pathophysiological overlap between depressive disorders (DDs) and metabolic disorders (MDs). Green: common pathophysiology of civilization diseases. * co-existence of DDs and MDs; HPA: hypothalamic-pituitary-adrenal.

**Table 1 life-11-00117-t001:** Depressive disorders in ICD-11.

6A70 Single episode depressive disorder
6A71 Recurrent depressive disorder
6A72 Dysthymic disorder
6A73 Mixed depressive and anxiety disorder
6A7Y Other specified depressive disorders
6A7Z Depressive disorders, unspecified

**Table 2 life-11-00117-t002:** The International Diabetes Federation (IDF) criteria of MetS [1].

● Central obesity (defined as waist circumference * with ethnicity specific values; for Caucasian race: Male ≥ 94 cm Female ≥ 80 cm);
**plus any two of the following four factors:**
● Raised triglycerides ≥ 150 mg/dL (1.7 mmol/L) or specific treatment for this lipid abnormality
● Reduced HDL cholesterol < 40 mg/dL (1.03 mmol/L) in males, < 50 mg/dL (1.29 mmol/L) in females or specific treatment for this lipid abnormality
● Raised blood pressure: systolic BP ≥ 130 or diastolic BP ≥ 85 mm Hg or treatment of previously diagnosed hypertension
● Raised fasting plasma glucose ≥ 5.6 mmol/L (100 mg/dL) or previously diagnosed type 2 diabetes

* If BMI is >30 kg/m^2^, central obesity can be assumed, and waist circumference does not need to be measured.

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
