# Peer review of "Adiposity in Depression or Depression in Adiposity? The Role of Immune-Inflammatory-Microbial Overlap"

_life, 2021, doi:10.3390/life11020117_

Round 1
Reviewer 1 Report
Minor comments
- There are some editing issues along the manuscript i.e. several expression are underlined. Please correct.
- Line 189 Much is know about the impact of dietary fibre, fish oil, or probiotics on gut-brain axis. Recently a high interest has been payed to dietary polyphenols, which are metabolised mainly by gut microbiota and they may influence their prevalence (Ceppa 2019 IJFSN). More importantly, some of the metabolites of polyphenols have been shown to pass BBB and thus directly influence brain-gut axis. I suggest authors to mention this information.
- Figure 1. As acronyms such as OxS and CLCGI are not common I suggest to not use them on the figure but spell them out.
Author Response
- There are some editing issues along the manuscript i.e. several expression are underlined. Please correct.
I corrected these issues. - Line 189 Much is know about the impact of dietary fibre, fish oil, or probiotics on gut-brain axis. Recently a high interest has been payed to dietary polyphenols, which are metabolised mainly by gut microbiota and they may influence their prevalence (Ceppa 2019 IJFSN). More importantly, some of the metabolites of polyphenols have been shown to pass BBB and thus directly influence brain-gut axis. I suggest authors to mention this information.
I mentioned the information in lines 178-181. - Figure 1. As acronyms such as OxS and CLCGI are not common I suggest to not use them on the figure but spell them out.
I did it.
Reviewer 2 Report
The manuscript is a comprehensive review which focuses on a clinically important issue. The relevant findings are discussed in a logical order; however, the text should be divided into several chapters. One important missing point in this review is that the Author did not mention how the appearance or function of brown and beige thermogenic adipocytes could be affected in obesity related to depressive disorders. The abbreviations CRP (lane 99) and SCFA (lane 151) should be defined and grammatical corrections (e.g. lanes 9, 28, 77, 189) should be made.
Author Response
- The manuscript is a comprehensive review which focuses on a clinically important issue. The relevant findings are discussed in a logical order;
however, the text should be divided into several chapters.
I divided the text into chapters. - One important missing point in this review is that the Author did not mention how the appearance or function of brown and beige thermogenic adipocytes could be affected in obesity related to depressive disorders.
I did a literature search and decided to add several pieces of information on BAT in different parts of the manuscript: lines 59-60, 89-97, 143-144, 149-150. - The abbreviations CRP (lane 99) and SCFA (lane 151) should be defined and grammatical corrections (e.g. lanes 9, 28, 77, 189) should be made.
I defined the abbreviations and tried to make grammatical corrections. - English language has been proofread twice by a native-speaker working for Medical University.
Thank you for all the comments.
Sincerely,
Oliwia Gawlik-Kotelnicka.
Round 2
Reviewer 2 Report
The manuscript was substantially improved and can be accepted for publication. The size of Figure 1 should be reduced.
Author Response
Thank you.
I reduced the size of Figure 1.